# Long-Term Effect of Modified Glass Ionomer Cement with Mimicked Biological Property of Recombinant Translationally Controlled Protein

**DOI:** 10.3390/polym14163341

**Published:** 2022-08-16

**Authors:** Prawichaya Sangsuwan, Wilaiwan Chotigeat, Sissada Tannukit, Ureporn Kedjarune-Leggat

**Affiliations:** 1Molecular Biology and Bioinformatics Program, Biological Science Division, Faculty of Science, Prince of Songkla University, Hat Yai, Songkhla 90112, Thailand; 2Department of Oral Biology and Occlusion, Faculty of Dentistry, Prince of Songkla University, Hat Yai, Songkhla 90112, Thailand; 3Cell Biology and Biomaterials Research Unit, Faculty of Dentistry, Prince of Songkla University, Hat Yai, Songkhla 90112, Thailand

**Keywords:** translationally controlled protein, human dental pulp stem cells, glass ionomer cement, tricalcium phosphate, regenerative

## Abstract

This study modified glass ionomer cement (GIC) by adding mimicked biological molecules to reduce cell death. GIC was modified to BIOGIC by adding chitosan and bovine serum albumin for enhancing protein release. The BIOGIC was supplemented with tricalcium phosphate (TCP) and recombinant translationally controlled tumor protein (TCTP) to improve its biological properties. Four groups of materials, GIC, BIOGIC, BIOGIC+TCP, and BIOGIC + TCP + TCTP, were examined by XRD and SEM-EDX. TCTP released from the specimens was determined by an ELISA method. Human dental pulp stem cells (hDPSCs) were harvested and analyzed by MTT assay, apoptosis, gene expression, and cell differentiation. All groups had the same crystallization characteristic peaks of La_2_O_3_. The elemental compositions composed of La, Si, and Al are the main inorganic components. The results show that BIOGIC + TCP + TCTP presented significantly higher percentages of cell viability than other groups on day 1 to day 23 (*p* < 0.05), but were not different after day 24 to day 41 and had reduced cell apoptosis including BAX, TPT1, BCL-2, and Caspase-3. The BIOGIC + TCP + TCTP demonstrated higher odontoblast mineralization and differentiation markers including ALP activity, DSPP, DMP-1, ALP, BMP-2, and OPN. It enhanced cell proliferation and differentiation as well as mineralization with down-regulation of genes related to apoptosis compared with other groups.

## 1. Introduction

The current concept of regenerative dentistry leads to a lot of interest in developing new restorative materials. Dental biomaterials mimicking biological strategies, including reducing cell death and promoting hard tissue regeneration, are required for this concept [1]. A tooth experiencing severe loss of its structure from aggressive conditions, such as deep caries that nearly expose the pulp, may require a dental lining material with a protective or regenerative property (see in Figure 1) [2].

Glass ionomer cement (GIC) has been broadly used in dental restorative and lining materials [3]. The elements of the glass component include F, Na, P, K, Cl, Si, Al, Ca, and La [4,5]. In one study, the conventional glass ionomer powder was mixed with a liquid mostly composed of polyalkenoic acids, mainly poly (acrylic acid) [6], being formed with an acid–base reaction. The important properties of GIC include long-term fluoride release [7,8], good adhesion to tooth structure, low cytotoxicity, and good biocompatibility [9,10]. It is also used in other medical fields for bone cement in orthopedic surgery, such as ear ossicles [11] and bone substitute plates for craniofacial reconstruction [12]. There have been many studies that modify GIC for some specific properties. For example, the modification with chitosan solutions with the addition of l-arginine in GIC to increase the antibacterial potential [13,14], and enhance the mechanical properties of GIC with cellulose microfibers [6]. Recently, there was a study that incorporated casein phosphopeptide-amorphous calcium phosphate (CPP-ACP) into GIC in order to enhance antimicrobial and remineralization properties [7]. Some studies reported that GIC supplemented with chitosan and bovine serum albumin (BSA) can enhance and prolong release of added protein [9] with its retained bioactivity [10].

Translationally controlled tumor protein (TCTP) or fortilin is a conserved multifunctional protein that is involved in many cellular activities (see review [15]) such as cell growth/proliferation [16], calcium (Ca^2+^) [17] scavenging, oxidative stress [18], cell cycle progression, protection of cells from various stress conditions, and anti-apoptosis/prosurvival [19]. Research conducted by our group showed that TCTP can reduce apoptosis from 2-hydroxy-ethyl methacrylate (HEMA)-treated pulp cells [20]. A study found that chitosan-fluoroaluminosilicate resin-modified glass ionomer cement supplemented with TCTP reduced apoptotic cells and promoted pulp cell mineralization [21].

Tricalcium phosphate (TCP), especially beta-tricalcium phosphate, is a compound that has been used to enhance mineralization or osteogenesis in many biomaterials [22,23]. However, there is no study on TCP and GIC. This study developed the new material based on GIC with enhanced regenerative biological properties for being a sub-lining or lining cement for a deep cavity restoration. By using mimicked biological molecules, a recombinant *Penaeus merguiensis* TCTP (*Pmer*-TCTP) was produced to use in this study. This protein comprised 168 amino acid residues, and the molecular mass was 19.2 kDa from *Penaeus merguiensis* shrimp [24]. This new glass ionomer powder was modified with added chitosan and albumin for increasing the release of *Pmer*-TCTP of the cement [25]. TCP was also added to the powder in order to activate cell differentiation and mineralization. The hypothesis of the study is that this modified GIC has better biocompatibility and higher mineralization capability than GIC, while the chemical structures are not much different. The objectives of this study were to investigate some chemical structures of the modified GICs and compare in vitro long-term effects of these specimens on dental pulp stem cells.

## 2. Materials and Methods

### 2.1. Ethical Statement

The pulp tissue required written informed consent from the participants, which was approved by the Human Research Ethics Committee of the Faculty of Dentistry, Prince of Songkla University, number code: EC 6211-047.

### 2.2. Cell Culture and Characterization of Human Dental Pulp Stem Cells (hDPSCs)

Sound human third molars were collected from three adults aged 18–25 years at the Dental Hospital, Faculty of Dentistry, Prince of Songkla University. The culture of pulp cells was produced using an enzymatic method [20]. Briefly, the pulp tissue was minced into pieces and digested in a solution of collagenase Type I and dispase (GIBCO^®^, Life Technologies Corporation, Carlsbad, CA, USA), cells were cultured in alpha-modified Eagle’s medium (α-MEM) supplemented with 10% FBS (GIBCO^®^, Life Technologies Corporation, Carlsbad, CA, USA), 0.05 mM L-ascorbic acid 2-phosphate (Sigma, Life Science, St. Louis, MO, USA), 100 µM L-glutamate, and 100 µg/mL amphotericin B (GIBCO^®^, Life Technologies Corporation, Carlsbad, CA, USA). The immunophenotypic characteristics of the mesenchymal stem cells (MSCs) of hDPSCs were examined according to the International Society for Cellular Therapy (ISCT) protocols [26]. Briefly, hDPSCs were treated with a pre-conjugated antibody using a Human MSC Analysis kit (Beckman Coulter, Villepinte, France) following the manufacturer’s instructions, and the samples were analyzed by a flow cytometer using CytExpert software (Beckman Coulter, Villepinte, France, Life Sciences, St. Louis, MO, USA, USA). The self-renewal capacity of hDPSCs was determined by the colony-forming unit fibroblast (CFU-F) assay. The hDPSCs were seeded at 100 cells/well. After 10 days, cells were fixed and stained with 0.4% crystal violet. The multipotent differentiation potentials of osteogenesis and adipogenesis of hDPSCs were investigated after osteogenic and adipogenic induction for 14 days. Alizarin red S and Oil Red O staining [27] were performed to detect the formation of mineral nodules and lipid droplets.

### 2.3. Expression and Purification of Pmer-TCTP Protein

The TCTP gene from *Penaeus merguiensis* (*Pmer*-TCTP) was expressed in the bacteria *Escherichia coli* (*E*. *coli*) strain BL21 method [25]. Briefly, the pGEX-*Pmer*-TCTP was inoculated and induced by IPTG (isopropyl β-D-thiogalactopyronositol) (Amresco^®^ Life Science, Lund, Sweden). Glutathione Sepharose 4 Fast Flow (GE Healthcare Bio-Science, Piscataway, NJ, USA) was used to purify the soluble protein and a GST-tagged protein was cut by thrombin (Amersham Biosciences, Piscataway, NJ, USA). A single band of TCTP protein molecular mass 19.2 kDa was detected by 12% SDS-PAGE and a Western blot assay using rabbit anti-*Pmer-*TCTP antibody (1:10,000) and goat–rabbit IgG conjugated with HRP (1:5000) (Jackson Immuno Research Laboratories, West Grove, PA, USA), respectively. A BCA protein assay kit (Pierce Biotechnology, Rockford, IL, USA) was used to determine the concentration of protein.

### 2.4. Preparation of Specimens

The conventional type of glass ionomer cement (GIC) 3M™ (3M ESPE ketac™ Molar Easymix, St. Paul, MN, USA) was used in this study. The powder (batch no. 632442) was composed of Al-Ca-La fluorosilicate glass and 5% copolymer acid (acrylic and maleic) and the liquid (batch no. 641902) components comprised polyalkenoic acid, tartaric acid, and water. Table 1 shows the composition of specimens in each group. The ratio by weight of powder:liquid of the GIC group was 4.5:1 following the manufacturer’s instructions and 1.17:1 in the other BIOGIC groups. The physical and biological test is summarized in Figure 2.

### 2.5. X-ray Diffraction (XRD)

X-ray diffraction (XGT 5200WR, Horiba, Japan) was used to identify diffraction patterns obtained from the specimens with the Joint Committee on Powder Diffraction Standards (JCPDS) database. A fixed position of 1 was set for each specimen. The detector was scanned between 25 and 60 angles. A step size of 0.02° was used, with a step time of 2.5 s.

### 2.6. Scanning Electron Microscopy (SEM) and Energy Dispersive X-ray Analysis (EDX)

The appearance of surface morphology and elements of the specimens were examined with a SEM with EDX (JSM-5800, Jeol, Japan). Each specimen was dehydrated in a graded ethanol–water series and mounted on aluminum stubs coated with gold.

### 2.7. Determination of the Released Pmer-TCTP

The recombinant *Pmer*-TCTP released from groups GIC + TCTP and BIOGIC + TCP + TCTP was measured by a direct ELISA method. Each specimen was placed in a 15 mL tube with 1 mL of PBS pH 7. 4 at 37 °C and shaken at 100 rpm/min. The immersed water was kept and refreshed every two days from day 1, 3, to 25. The immersed PBS was lyophilized and suspended in 100 µL of a coating buffer (0.1 M Na_2_CO_3_, 0.1 M NaHCO_3_ pH 9.6) before the direct ELISA assay. At room temperature, 100 µL of each sample was spread over an ELISA plate and incubated overnight. The plate was washed six times with 200 µL of PBST (10 mM phosphate buffer pH 7.4 with 0.05% Tween 20). The remaining protein-binding site was blocked by adding a 200 µL blocking buffer (5% non-fat dry milk in PBST) each well and incubated for 2 h at room temperature. The plate was washed and incubated with 100 µL (diluted 1:1000 in blocking buffer) of anti-TCTP antibody (Abcam^®^, Cambridge, UK) per well and kept at 4 °C overnight. Then, the plate was washed and 100 µL of anti-mouse-alkaline phosphatase (AP) (Thermo Fisher Scientific Inc., Rockford, IL, USA) (diluted at 1:5000 in a blocking buffer) was added and incubated for 2 h at room temperature. The plate was washed again and each well was added with 100 µL of the substrate solution (pNPP; p-nitrophenyl-phosphate). After color development, 100 µL of stop solution (1 N NaOH) was added. The absorbance was read at 405 nm using a microplate reader (Biochrom Anthos Zenyth 200rt, Holliston, MA, USA). The amount of *Pmer*-TCTP in each sample was calculated from the standard curve of known concentrations of *Pmer*-TCTP.

### 2.8. Long-Term Cell Viability Assay

We assessed the long-term effect of different test specimens on hDPSC viability after the release of substances from the specimens exposed to cells at 21 periods of time; the first period lasted 24 h and the others lasted 2 days. The hDPSCs were seeded on the lower part of 24-well Transwell^®^ (Costar^®^, Corning., Life Science, St. Louis, MO, USA, USA) polystyrene-coated plates (6.5 mm insert with 8.0 µm pore size polycarbonate membrane) and were incubated at 37 °C in a humidified atmosphere containing 5% CO_2_. After 24 h, each specimen from 4 groups of GICs (*n* = 6 in each group) was placed on the insert part. After 24 h (day 1), MTT assay was performed [28], and each specimen was moved to another new transwell insert that had already been seeded with 2.0 × 10^4^ hDPSCs for 24 h. Every two days from day 3 to 41, cell viability was determined by MTT assay. The optical density (OD) was corrected for a blank (medium only), divided by the OD of the control (cell culture without specimen), and expressed as a percentage of the control, which represented the percentage of cell viability.

### 2.9. Apoptotic Assay

Cell apoptosis was investigated with hDPSCs cultured on specimens by flow cytometry using a propidium iodide (PI) and annexin V-FLUOS double staining kit (Roche Diagnostics Corporation, Roche Applied Science, Indianapolis, IN, USA). The samples were analyzed by a flow cytometer and the data were calculated using guavaSolf^TM^ version 2.7 software (Guava easyCyte^TM^ HT, Hayward, CA, USA).

### 2.10. Alkaline Phosphatase (ALP) Activity Assay

The hDPSCs were seeded onto each specimen and cultured with osteogenic medium containing 10 mM beta-glycerophosphate, 100 units/mL penicillin, 100 mg/mL amphotericin B, 0.05 mM L-ascorbic acid 2-phosphate, and 100 mM Dexamethasone in α-MEM with 10% inactivated FBS. After 24 h, the medium was replaced with new medium every two days for 7, 14, 21, 28, 35, and 42 days. The ALP activity was measured using *p*-nitrophenol phosphate in 0.1 M 2-amino-2-methyl-1-propanol (AMP), 2 mM MgCl_2_, and pH 10.5 as a substrate. The reaction was left for 30 min at 37 °C and stopped by adding 0.8 mL of 50 mM NaOH to each well. The absorbance was measured at 405 nm (Biochrom Anthos Zenyth 200 rt, Holliston, MA, USA) against its own blank. The total protein of each sample was determined by a BCA kit (Pierce Biotechnology, Holmdel, NJ, USA). The ALP activity was calculated as *p*-nitrophenol per µg of total protein and presented as µmole/min/µg of protein.

### 2.11. Alizarin Red Staining (ARS)

hDPSCs seeded onto each specimen and cultured in osteogenic medium. On days 7, 14, 21, 28, 35, and 42, the ARS assay was performed. Briefly, the medium was removed and washed with PBS pH 7.4. Cells were fixed with formaldehyde and stained with 40 mM of Alizarin red S (ARS). Calcium deposition was dissolved by cetylpyridinium chloride (CPC) solution. The absorbance of the samples was measured with a microplate reader of 550 nm against its own blank.

### 2.12. Determination of Gene Expression by Reverse Transcription Quantitative Polymerase Chain Reaction (RT-qPCR)

The step of gene expression (RT-qPCR) was 95 °C for 10 min, followed by 45 cycles of denaturation at 95 °C for 30 s, annealing at 58 °C for 30 s, and elongation at 72 °C for 45 s. The hDPSCs were seeded at 1.0 × 10^5^ cells onto each specimen and the control group was cells cultured with normal medium only. After 24 h, the medium was refreshed, then replaced every two days. The RT-qPCR was performed on days 7, 14, 21, 28, 35, and 42. The total cellular RNA was extracted and purified using PureLink™ RNA Mini Kit (Invitrogen, Burlington, ON, Canada) and then reverse transcribed using the SuperScript III First-Strand Synthesis System (Invitrogen, Burlington, ON, Canada). Reverse transcription quantitative polymerase chain reaction (RT-qPCR) was performed using the SensiFAST™ SYBR^®^ No-ROX Kit (Bioline, London, UK) and the Light Cycler system (Roche Diagnostics, Mannheim, Germany), and the specific primers were designed according to the cDNA sequences from GenBank (see Table 2). The expression was analyzed with GAPDH serving as a housekeeping/reference gene for normalization. Relative expression levels were calculated using the 2^−∆∆CT^ [29].

### 2.13. Statistical Analysis

The data are shown as means ± standard deviations (SD), and the data were examined for normal distribution with the Shapiro–Wilk test. Two-way repeated analysis of variance (ANOVA) was used to analyze the results of TCTP release. One-way ANOVA and Tukey’s post hoc comparison test were used to investigate the differences between groups in MTT assay, apoptosis by flow cytometry, ARS assay, and RT-qPCR. Statistically significant difference was set at *p* < 0.05.

## 3. Results

### 3.1. Characterization of hDPSCs

Some of the isolated hDPSCs were capable of differentiating into multiple lineages, which is one of the key properties of MSCs. The differentiation potential of hDPSCs was evaluated by culturing the cells in osteogenic and adipogenic medium to confirm their differentiation capacity. The number of colonies from the low-density culture that were stained with crystal violet revealed their self-renewal capacity (see Figure 3C). This result demonstrated that the cells had capacity to generate new colonies from single cells. Figure 1A shows condensed nodules of calcium stained with Alizarin Red that were sparsely scattered throughout the adherent layer as single mineralized zones. Adipogenic differentiation was indicated by the presence of intracellular lipid droplets, which were shown by Oil Red O stained (see Figure 3B). The morphology of certain cells changed from spindle-like to polygonal shapes. The levels of surface marker expression from expanded were determined using flow cytometry. The cells were extensively expressed for specific MSC markers CD73 (99.80%), CD90 (97.86%), and CD105 (98.18%), while the non-expression levels of hematopoietic stem cell markers were CD34 (0.05%) and CD45 (0.00%) (see Figure 3D). These results confirmed that the isolated cells used in this study were dental pulp stem cells.

### 3.2. X-ray Diffractometer (XRD)

Parallel X-ray diffraction (XRD) measurements were carried out to determine the structure of different GICs and are shown in Figure 4. They all have similar patterns, especially the crystallization characteristic peaks of La_2_O_3_.

### 3.3. Energy-Dispersive X-ray Spectroscopy (EDX)

The SEM-EDX spectra of different GICs are presented in Figure 5. All specimens presented similar patterns and sharp peaks of typical GIC. Elemental analysis showed that La, Si, and Al are the main inorganic components, which were lower in BIOGIC groups and replaced by the increase in O and C that came from the compositions of chitosan and protein.

### 3.4. Pmer-TCTP Release

The results of measuring the released concentrations of *Pmer*-TCTP from different specimens using ELISA are shown in Figure 6. GIC was set as negative control, which did not detect *Pmer-*TCTP. Both specimens, BIOGIC + TCP + TCTP and GIC + TCTP, had similar patterns of release, which reached the highest concentration on day 1, and the protein release was gradually decreased from day 3 to day 15. Subsequently, the protein content was not detectable by ELISA. Using two-way repeated ANOVA found that BIOGIC + TCP + TCTP released *Pmer-*TCTP at significantly (*p* < 0.01) higher concentrations from day 1 to day 15 than GIC + TCTP. There was no interaction between groups of the material and time of release.

### 3.5. Long-Term In Vitro Effect of Different GICs on hDPSCs

From Figure 7, BIOGIC + TCP + TCTP had higher percentages of cell viability that were significantly higher than GIC, from day 1 to day 23 except on day 13 and 19. In addition, the percentages of cell viability in this group were over 100% on day 5, 7, 9, and 11, which suggested that this specimen can promote cell proliferation at those periods. It was noticed that there was no significant difference between the percentages of cell viability in all groups from day 25 to the end of the study.

### 3.6. Apoptosis Assay

Flow cytometry was performed to analyze hDPSC apoptosis after being cultured onto different types of GICs for 24 h (see Figure 8). Cells cultured with medium only were set as control. The percentage of cell apoptosis (positive to PI and annexin-V) in the GIC group was significantly (*p* < 0.05) higher than other groups. The percentages of early apoptosis (positive to annexin-V only) in BIOGIC + TCP + TCTP and GIC were significantly higher than other groups. However, there was no significant difference between the percentages of living cells in all groups, which were 80%. These results indicated that none of the specimens had high cytotoxicity that caused cell apoptosis.

### 3.7. Alkaline Phosphatase (ALP) Activity

The ALP activity of hDPSCs after being cultured on different types of GIC specimens is shown in Figure 9. The ALP activity of cells in the BIOGIC + TCP + TCTP group was significantly (*p* < 0.01) higher than other specimen groups on day 7, 14, 21, and 28. The ALP activity of cells in this group was highest on day 7, after which they gradually decreased until day 42. However, the ALP activity of cells on GIC was lower than the other groups, especially on day 35 and 42.

### 3.8. Mineralization of hDPSCs

The results of the calcium deposition (see Figure 10) show that BIOGIC + TCP and BIOGIC + TCP + TCTP had significantly (*p* < 0.01) higher calcium content compared to GIC after being cultured for 14 days. In addition, the BIOGIC + TCP + TCTP group, after being cultured for 42 days, was significantly (*p* < 0.01) higher than other groups including the control. Cells being cultured on GIC had lower calcium content compared to the other groups. It was noticed that the control group, which features hDPSCs cultured with osteogenic medium without any specimen, had the highest calcium deposition than other groups at the same time period on day 7, 14, and 21 and it reached the highest calcium deposition on day 28.

### 3.9. Apoptosis-Related Gene Expression by qRT-PCR

The expression of pro-apoptosis (BAX, Caspase-3) and anti-apoptosis (TPT1 and BCL-2) was analyzed by qRT-PCR as shown in Figure 11. The highest level of BAX expression was observed in the GIC group on day 1 and day 3 (*p* < 0.05). The expression of the BAX gene in the BIOGIC + TCP + TCTP group was not higher than GIC specimens in most of the time periods. Moreover, down-regulation was observed on day 35 and day 42. Like BAX mRNA expression, the expression of Caspase-3 (see Figure 11C) of cells cultured on most specimen groups was up-regulated on day 1. Then, the expression was reduced and down-regulated compared with controls in all specimens from day 28 to day 42. The pattern of the expression of TPT1 shows that all specimens were up-regulated on day 1 and day 3 and down-regulated from day 5 to day 42 (see Figure 11B). However, the BIOGIC + TCP + TCTP group had significantly (*p* < 0.05) lower mRNA expression than the GIC group on day 1 and day 3. Moreover, from day 5 to day 42, the TPT1 of cells in BIOGIC + TCP + TCTP was down-regulated. The expression of BCL-2 mRNA (see Figure 11D) did not alter much in most specimens. In the BIOGIC group, the BCL-2 gene expression was higher than BIOGIC + TCP + TCTP at day 1. BCL-2 gene was down-regulated in the GIC group at day 42.

### 3.10. mRNA Gene Expression Related to Odontogenic/Osteogenic Differentiation

The response of the DSPP gene was reported as an average folds expression compared to the control group (see Figure 12A). On day 7, the expression of the DSPP mRNA in cells cultured on BIOGIC and BIOGIC + TCP + TCTP specimens was up-regulated more than the control, then the expression slowed down. The hDPSCs cultured on BIOGIC + TCP + TCTP had higher DSPP expression than other specimens on day 7. The expression patterns of the ALP gene (see Figure 12C) had six-fold up-regulation on day 14, then down-regulation. This ALP mRNA expression corresponded to the result of ALP activity. The BIOGIC + TCP + TCTP group had significantly (*p* < 0.05) higher up-regulation than other groups on day 7 and day 14. DMP-1 expression was significantly down-regulated (see Figure 12B) in all groups after being cultured from day 1 to day 7. After that, the up-regulation that was found, especially in hDPSCs cultured with BIOGIC + TCP + TCTP, was significantly (*p* < 0.05) higher than other groups from day 14 to 42. BMP-2 gene expression reduced over time in all groups (see Figure 12D) on day 1, 3, 5, 7, and 14 compared to the control, which was similar to DMP-1. However, cells in BIOGIC + TCP + TCTP had higher expression than other groups on day 21 to 42. The expression of the DMP-1 pattern was similar to BMP-2, which was down-regulated in the early phase (see Figure 12D). BIOGIC + TCP + TCTP demonstrated the highest up-regulation after being cultured on day 28, 35, and 42. The pattern of the expression of OPN was significantly (*p* < 0.05) up-regulated on day 5 (see Figure 12E) in cells cultured on BIOGIC + TCP + TCTP, followed by down-regulation.

## 4. Discussion

The modified glass ionomer cement used in this study aimed to enhance regeneration of dental tissue as a deep lining cement in tooth restoration. The new material should release biological molecules that can reduce cell death and activate the remineralization process. By mimicking the biological property of recombinant *Pmer*-TCTP, the conventional glass ionomer cement was modified with chitosan BSA and TCP. This study used pulp stem cells, hDPSCs, to evaluate long-term in vitro effect of the materials. The hDPSCs have MSC characteristics, including high clonogenic capacity, rapid proliferation, and multipotential differentiation, that are able to differentiate, especially odontoblasts [30].

The results from XRD and EDX revealed that the main structure of the modified specimens, BIOGICs, still has the same pattern as GIC with crystallization characteristic peaks of La_2_O_3_ with the increase in O and C components. The results of ELISA assay revealed that BIOGIC + TCP + TCTP can release higher amounts of TCTP than GIC + TCTP, which corresponds with previous studies [9,10,25]. A recent study [25] revealed that GIC with added chitosan, albumin, and TCTP can release a higher amount of TCTP than GIC + TCTP. That study evaluated only 24 h release and there was no TCP added, but it evaluated the release for 25 days and the released TCTP was detected until day 15. Thus, the present study found that the added TCP did not affect the release of TCTP. It has been suggested that chitosan can form a polyelectrolyte complex with poly acrylic acid, the main components of the liquid part of glass ionomer cement, and can prolong the release of the active molecules [31,32].

The MTT assay results revealed that all three BIOGICs did not have high cytotoxicity, although they could promote cell growth on day 5 to 17. This confirmed that chitosan, BSA, TCP, and TCTP were not cytotoxic; in particular, TCTP can reduce cell death, as shown in the apoptosis assay. However, none of the specimens had a cytotoxic effect on hDPSCs after day 31.

The up-regulation of pro-apoptotic genes, BAX, and Caspase-3 on day 1 and 3 corresponded with the results of MTT assay. However, BIOGIC + TCP + TCTP down-regulated these genes on day 5, which may come from the reduced cytotoxicity and the properties of *Pmer*-TCTP. The up-regulation of two anti-apoptotic genes, TPT1 and BCL-2, at the early phase of all specimens may be due to cellular stress. These two genes were down-regulated from day 5 to 42 in all specimens, which may be due to the reduced cytotoxicity, especially the apoptotic effect of the specimens.

The main objective to supplement recombinant *Pmer*-TCTP into BIOGIC is to reduce apoptotic cells that may result from bacteria byproducts occurring during the development of caries and may also occur during deep cavity preparation and restoration [33,34]. The property of this specimen needs further investigation, especially in a caries model or animal study. However, the results of this study revealed that the BIOGIC + TCP + TCTP can up-regulate ALP mRNA expression in the early phase followed by increased ALP activity and mineralization higher than other specimens. This bioactivity has also been reported in other studies [25,35]. It was noticed that TCP in BIOGIC + TCP can also promote mineralization. It had higher calcium deposition than the BIOGIC and GIC groups (see Figure 8). TCP is one component in biomaterials that has been used for hard tissue engineering or bone regeneration [36,37].

The results of mRNA of genes related to odontogenic/osteogenic differentiation of hDPSCs are interesting. At an early phase (day 1–7), hDPSCs cultured onto all specimens up-regulated DSPP, ALP, and OPN expressions. DSPP is a marker of odontoblast differentiation and ALP is the early marker of mineralization. Later, on day 28–42, there was up-regulation of DMP-1 and BMP-2, especially in the BIOGIC + TCP + TCTP group. DMP-1 is a non-collagenous matrix protein necessary for the maturation of odontoblasts, as well as for mineralization [38]. BMP-2 plays a role in mineralization [39] and regulated DSPP expression in odontoblasts [40]. Osteopontin (OPN) is a highly phosphorylated glycoprotein that is a prominent component of the mineralized extracellular matrix of bone that was found at the calcification front of reparative dentin [41]. It could be suggested that this material group, BIOGIC + TCP + TCTP, can activate dental pulp stem cells to differentiate and promote the mineralization process.

## 5. Conclusions

The modified glass ionomer cement with added chitosan/albumin/TCP/TCTP has the same crystallization characteristic peaks of La_2_O_3_ as GIC. It can release a recombinant TCTP for at least 15 days. Compared to GIC, this material has less cytotoxicity and can promote hDPSCs growth and differentiation as well as mineralization. It could be used as a lining cement that enhances dentin regeneration; however, it requires further evaluation.

## Figures and Tables

**Figure 1 polymers-14-03341-f001:**
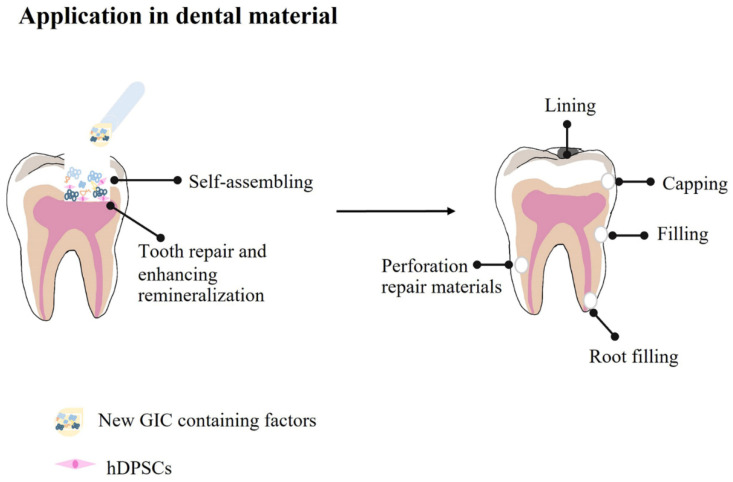
Conceptualization and utilization of new glass ionomer cement in clinical application.

**Figure 2 polymers-14-03341-f002:**
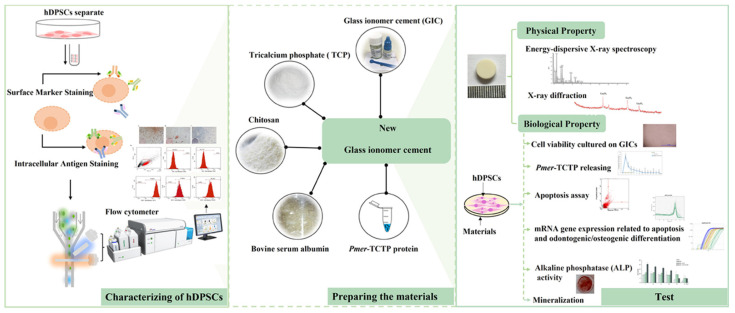
Overview schematic illustration of the physical and biological evaluation.

**Figure 3 polymers-14-03341-f003:**
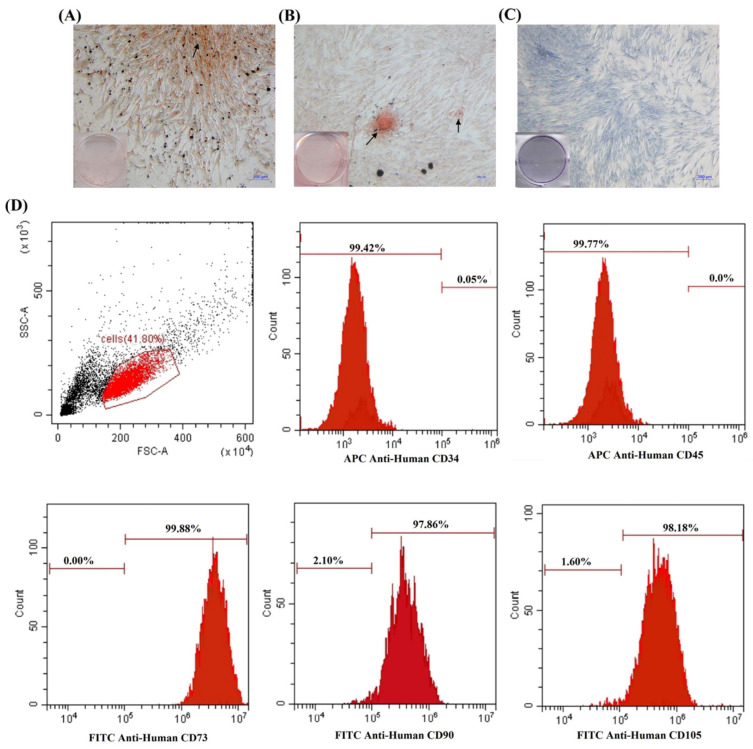
Isolation, morphological observation and flow cytometry analysis of hDPSCs. (**A**) Mineralized deposits identified by Alizarin Red staining in cells grown in osteogenic medium. (**B**) Adipogenic-induced cells were stained with Oil Red O. (**C**) Colony-forming fibroblastic units (CFU-F). (**D**) Histograms from flow cytometry analysis of cell surface markers. The hematopoietic marker, CD34 and CD45 positive are less than 20%. The MSC markers, CD73, CD 90, and CD105 positive are more than 80%.

**Figure 4 polymers-14-03341-f004:**
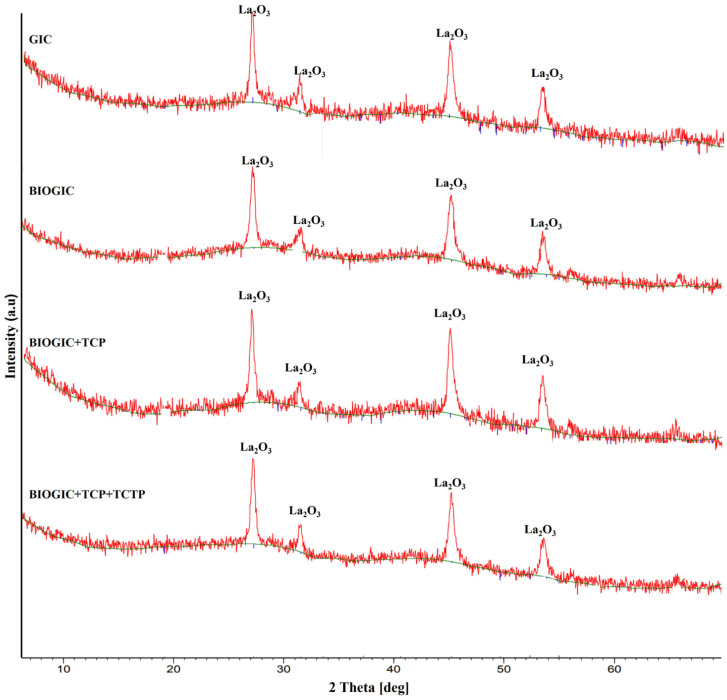
X-ray diffraction (XRD) patterns of GIC, BIOGIC, BIOGIC + TCP, and BIOGIC + TCP + TCTP.

**Figure 5 polymers-14-03341-f005:**
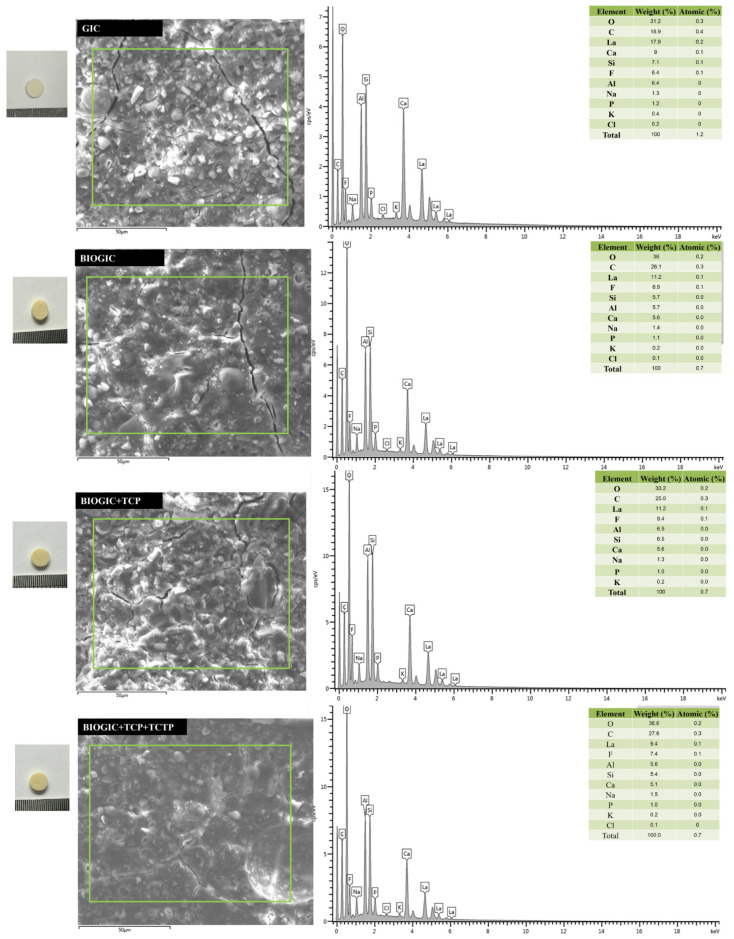
SEM-EDX spectrograms of GIC, BIOGIC, BIOGIC + TCP, and BIOGIC + TCP + TCTP found in the density fraction of b < 2.0 g cm^−3^ (cts = counts).

**Figure 6 polymers-14-03341-f006:**
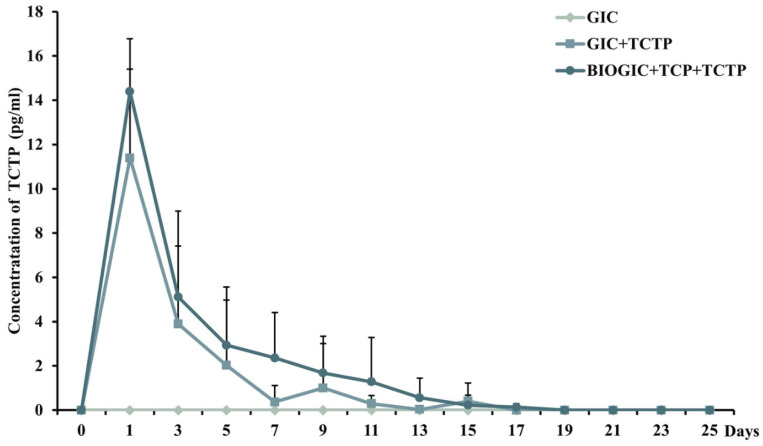
ELISA assay determined concentration of TCTP. TCTP was released from GIC and BIOGIC with added TCTP on day 1 to day 25.

**Figure 7 polymers-14-03341-f007:**
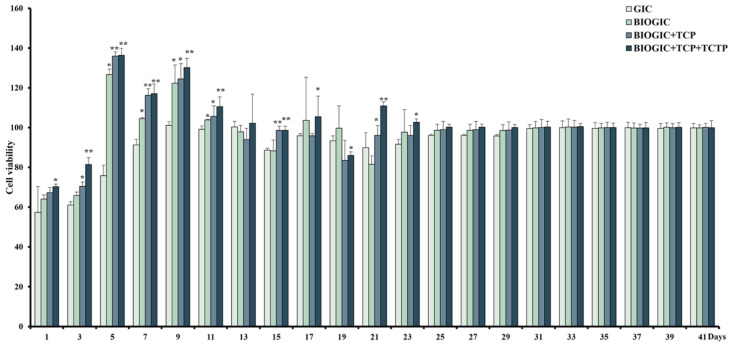
Long-term in vitro effect of different GICs on hDPSC viability. Results of MTT assay of cells cultured with different groups of GICs for 24 h to 41 days using transwell system. Data are represented as mean ± SD of cell populations. Value significant difference at * *p* < 0.05 and ** *p* < 0.01 compared to the GIC group (*n* = 6 in each group).

**Figure 8 polymers-14-03341-f008:**
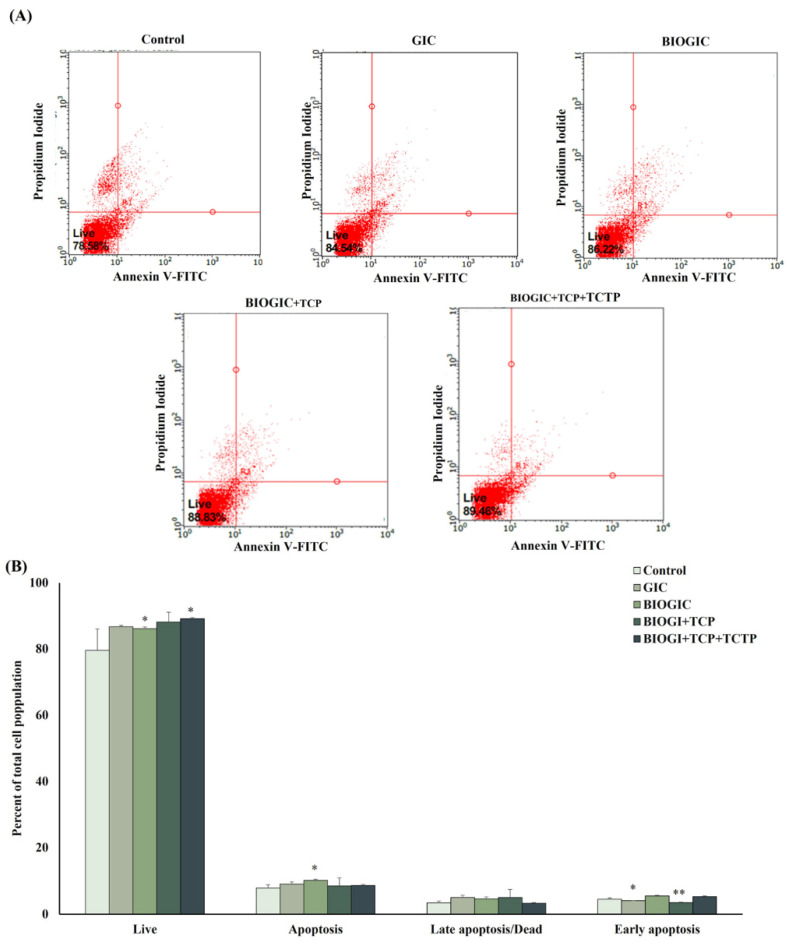
Anti-apoptotic property of TCTP against GIC. Cells were incubated with each group of specimens for 24 h. The apoptotic cells were determined by flow cytometry. (**A**) The flow cytometer images. (**B**) The percentage of total cell population. Data are represented as mean ± SD. Value significant difference at * *p* < 0.05 and ** *p* < 0.01 compared to control (*n* = 4 in each group).

**Figure 9 polymers-14-03341-f009:**
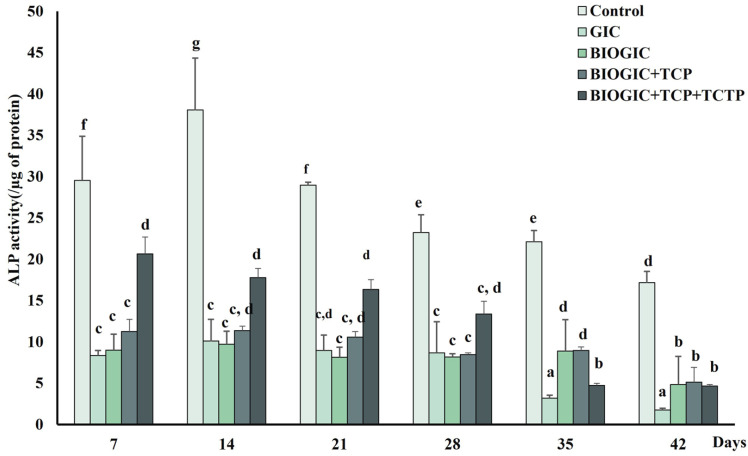
Odontogenic differentiation potential of hDPSCs. ALP activity was used to evaluate cell differentiation after culturing hDPSCs with each specimen for 7, 14, 21, 28, 35, and 42 days. Data are represented as mean ± SD. Different lowercase letters represent significant difference at *p* < 0.05 (*n* = 6 in each group).

**Figure 10 polymers-14-03341-f010:**
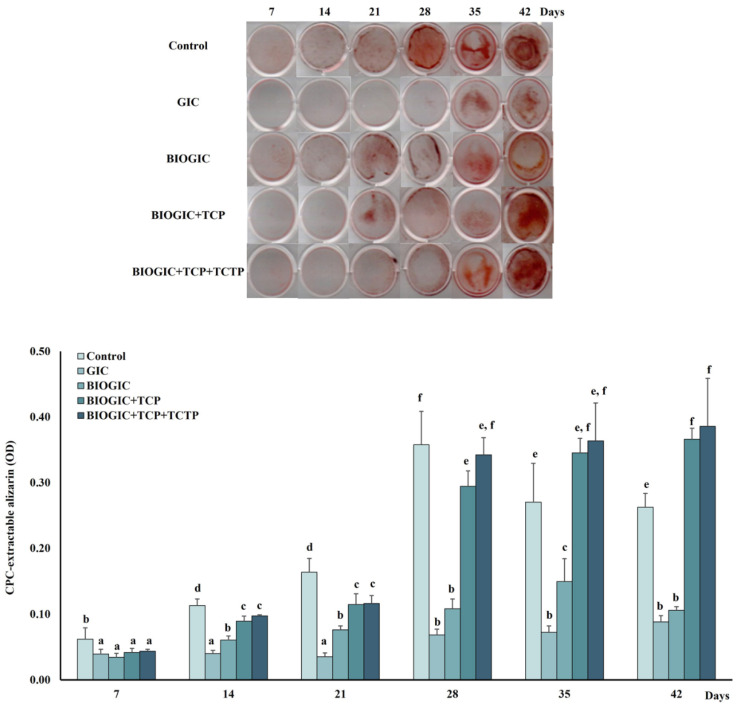
Odontogenic mineralization potential of hDPSCs. The mineralization was measured by alizarin red S staining, hDPSCs were cultured on the specimens for 7, 14, 21, 28, 35, and 42 days. Data are represented as mean ± SD. Different lowercase letters represent significant difference at *p* < 0.05, one-way ANOVA with Tukey multiple comparison test (*n* = 6 in each group).

**Figure 11 polymers-14-03341-f011:**
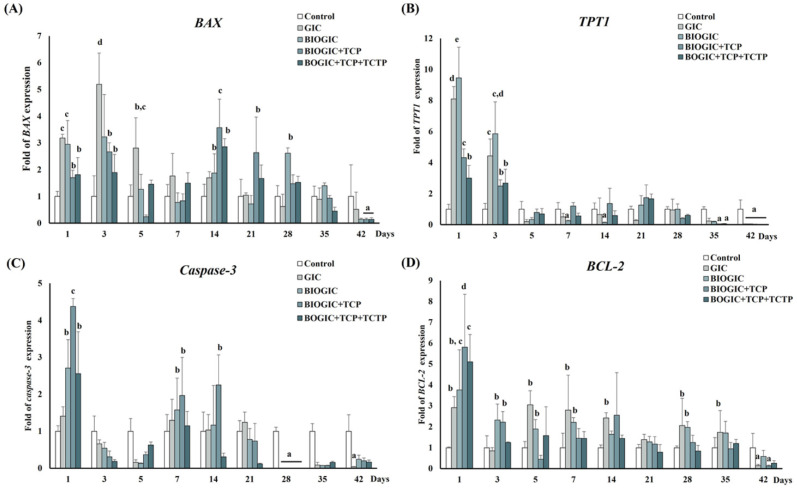
The quantification of mRNA gene expression related to cell apoptosis in hDPSCs cultured on each type of GIC for 1, 3, 5, 7, 14, 21, 28, 35, and 42 days was determined by qRT-PCR. (**A**) BCL2-associated X (BAX), (**B**) tumor protein translationally controlled 1 (TPT1), (**C**) (Caspase-3), and (**D**) B-cell lymphoma 2 (BCL-2). Data are represented as mean ± standard deviation (SD) of cell populations. Different lowercase letters represent significant difference between different groups within each time period at *p* < 0.05, one-way ANOVA with Tukey multiple comparison test (*n* = 4 in each group).

**Figure 12 polymers-14-03341-f012:**
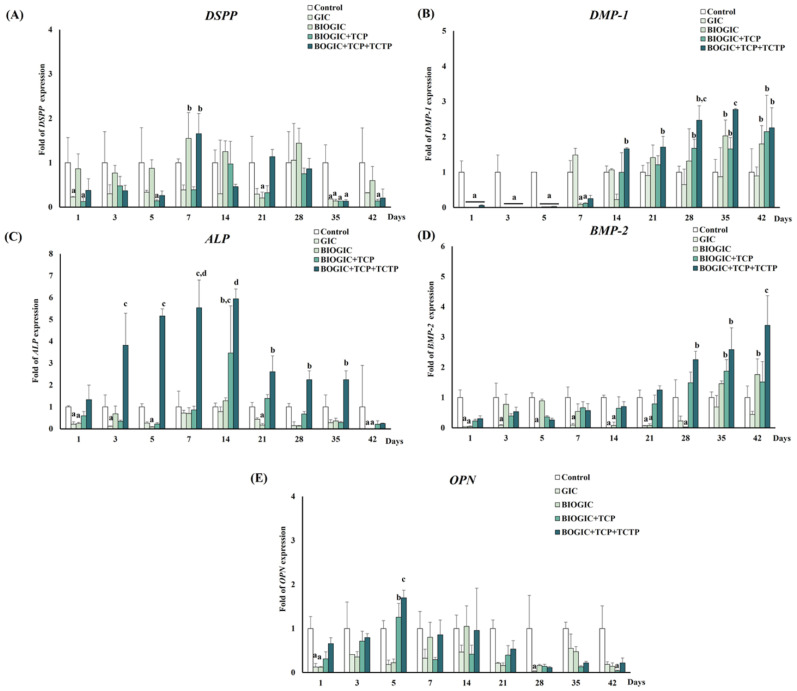
The quantification of the mRNA gene expression related to odontogenic/osteogenic differentiation in hDPSCs cultured on each type of GIC for 1, 3, 5, 7, 14, 21, 28, 35, and 42 days. (**A**) Dentin sialo phosphoprotein (DSPP), (**B**) dentin matrix protein 1 (DMP-1), (**C**) alkaline phosphatase (ALP), and (**D**) bone morphogenetic protein-2 (BMP-2), and (**E**) osteopontin (OPN). Data are represented as mean ± standard deviation (SD) of cell populations. Different lowercase letters represent significant difference at *p* < 0.05 (*n* = 4) between different groups within each time period (one-way ANOVA with Tukey multiple comparison test).

**Table 1 polymers-14-03341-t001:** Components of each cement powder.

Groups	Powder Compositions (% by Weight)
GIC	100% of glass ionomer cement powder
BIOGIC	80% of glass ionomer cement powder, 15% of chitosan, and 5% of bovine serum albumin
BIOGIC + TCP	79.95% of glass ionomer cement powder, 15% of chitosan, 5% of bovine serum albumin, and 0.05% of tricalcium phosphate
BIOGIC + TCP + TCTP	79.95% of glass ionomer cement powder, 15% chitosan, 5% of bovine serum albumin, 0.05% of tricalcium phosphate, and 1 µg *Pmer*-TCTP (added when mixing the powder and liquid components)

**Table 2 polymers-14-03341-t002:** The oligonucleotide primer sequences of qRT-PCR analysis *.

Gene	Primer Sequence (5′-3′)	GenBank Code
BAX	F: TGCTTCAGGGTTTCATCCAG R: GGCGGCAATCATCCTCTG	NC_000019
TPT1	F: AAATGTTAACAAATGTGGCAATTAT R: AACAATGCCTCCACTCCAAA	NM_003295.3
Caspase-3	F: AGAACTGGACTGTGGCATTGAG R: GCTTGTCGGCATACTGTTTCAG	NC_000004.12
BCL-2	F: TTTGAGTTCGGTGGGGTCAT R: TGACTTCACTTGTGGCCCAG	NM_000633
DSPP	F: GGGATGTTGGCGATGCA R: CCAGCTACTTGAGGTCCATCTTC	NM_014208.3
DMP-1	F: GCAGAGTGATGACCCAGAG R: GCTCGCTTCTGTCATCTTCC	NM_004407.3
ALP	F: CCACAAGCCCGTGACAGA R: GCGGCAGACTTTGGTTTC	NM_001127501
BMP-2	F: GCTTCCGCCTGTTTGTGTTTG R: AAGAGACATGTGAGGATTAGCAGGT	NM_007553
OPN	F: ACACATATTGATGGCCGAAGGTGA R: TGTGAGGTGATGTCCTCGTCTGT	NM_00582.2
GAPDH	F: GCACCGTCAAGGCTGAGAAC R: ATGGTGGTGAAGACGCCAGT	NM_001289745.1

* Primers were designed using the database from the National Center for Biotechnology Information (NCBI). F, forward; R, reverse. BCL2-associated X (BAX), tumor protein translationally controlled 1 (TPT1), Caspase-3, B-cell lymphoma 2 (BCL2), dentin sialo phosphoprotein (DSPP), dentin matrix protein 1 (DMP1), alkaline phosphatase (ALP), bone morphogenetic protein-2 (BMP-2), osteopontin (OPN), and glyceraldehyde-3-phosphate dehydrogenase (GAPDH).

## Data Availability

No data were used to support this study.

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
