# Peer review of "Long-Term Effect of Modified Glass Ionomer Cement with Mimicked Biological Property of Recombinant Translationally Controlled Protein"

_polymers, 2022, doi:10.3390/polym14163341_

Round 1

Reviewer 1 Report

The present manuscript is an interesting study. There are some issues in this manuscript that should be addressed as follows:

The novel aspects of this study should be mentioned in the introduction and the needs of this study. And the introduction past needs to be improved.

Why these compoments used in cement powder? There is no statement about the reasons chosen to be researched. Again missing the novelty.

The abstract needs to be re-written as well as conclusions section.

Please organize an abbreviation section to be easier for readers to follow.

The discussion should provide more details to analyze of the results of the present study, supported by relevant references.

The manuscript should be revised to improve the quality of the language. Many phrases are almost incomprehensible, or in others something is missing.

The manuscript should be checked regarding grammatical errors and plagiarism.

Author Response

Reviewer #1 comments: The present manuscript is an interesting study. There are some issues in this manuscript that should be addressed as follows:

1.The novel aspects of this study should be mentioned in the introduction and the needs of this study. And the introduction past needs to be improved.

Response: We have corrected the introduction especially try to explain about the novel aspects of the study.

2.Why these compoments used in cement powder? There is no statement about the reasons chosen to be researched. Again missing the novelty.

Response: The reasons about the components added in the powder were added in the introduction part. Chitosan and albumin was added in order to enhance the recombinant TCTP release and TCP was added to activate mineralization.

3.The abstract needs to be re-written as well as conclusions section.

Response: We have rewritten the abstract as suggested.

4.Please organize an abbreviation section to be easier for readers to follow.

Response: We have corrected as suggested.

5.The discussion should provide more details to analyze of the results of the present study, supported by relevant references.

Response: We added more information in the discussion part as suggested

6.The manuscript should be revised to improve the quality of the language. Many phrases are almost incomprehensible, or in others something is missing.

The manuscript should be checked regarding grammatical errors and plagiarism.

Response: We have corrected the language and grammatical errors as well as checked the plagiarism as suggested.

Reviewer 2 Report

Comments 0731

1.     Abstract: please to provide full name of hDPSCs and BIOGIC+TCP+TCTP.

2.     Introduction:                                                                  2.1 Current concept of regenerative dentistry ……..with protective or regenerative property (see in Figure 1). à please to provide reference(s).                                                                  2.2 Besides, it is used in other medical fields ………for craniofacial reconstruction. à please to provide reference(s).                                              2.3 (see review [11]) à [11] ?

2.4  please to provide the hypotheses of this study.

3.     Discussion:

3.1 BMP-2 plays a role in mineralization [34] and regulated DSPP expression in odontoblasts. à please to provide reference(s) about “BMP-2 regulated DSPP”.  

Author Response

Reviewer #2

  1. Abstract: please to provide full name of hDPSCs and BIOGIC+TCP+TCTP.

Response: We have corrected the abstract as suggested.

  1. Introduction:

2.1 Current concept of regenerative dentistry ……..with protective or regenerative property (see in Figure 1).

 à please to provide reference(s).

Response: The reference was added.

2.2 Besides, it is used in other medical fields ………for craniofacial reconstruction. à please to provide reference(s).

Response: The reference was added.

  2.3 (see review [11]) à [11] ?

Response: The reference was corrected.

2.4  please to provide the hypotheses of this study.

Response: The hypothesis of the study was added in the introduction part.

  1. Discussion:

3.1 BMP-2 plays a role in mineralization [34] and regulated DSPP expression in odontoblasts. à please to provide reference(s) about “BMP-2 regulated DSPP”.

Response: The references were corrected.

We hope that our responses satisfy the reviewers’ comments and suggestions, but let us know if we can assist further.

Round 2

Reviewer 2 Report

none